# UniTSGAN: A Unified Transformer-based Framework for Imbalanced Time Series Generation and Classification

## Abstract

Handling severe class imbalance in time series data is a critical challenge, especially for rare-event prediction in domains such as space weather forecasting and health monitoring. Standard discriminative models often perform poorly on the underrepresented minority class, which typically represents the most important outcomes for decision-making. Common remedies like oversampling or undersampling can cause overfitting or information loss. Generative Adversarial Networks (GANs) have shown promise in generating realistic synthetic data, but they are generally not optimized for class-discriminative generation and must be combined with separate classifiers. In this paper, we propose *UniTSGAN*, a unified adversarial framework that jointly handles multivariate time series generation and binary classification in highly imbalanced scenarios. Our model uses a transformer encoder for both the generator and discriminator. The generator can be pretrained with an unsupervised masking-based objective to learn latent representations of the minority class, and the discriminator has a dual-head architecture that simultaneously performs real-vs-fake discrimination and class label prediction. This design allows the model to learn realistic, class-consistent synthetic samples and a robust classifier in a single training process. To evaluate generative performance, we introduce a classification-based metric that measures how much adding synthetic data improves downstream classification. Experimental results on seven real-world datasets demonstrate that UniTSGAN consistently outperforms state-of-the-art methods on both imbalanced time series classification and generation tasks, particularly in low-data regimes.

## 1 Introduction

Time series data arise in domains such as healthcare, finance, and space science, where capturing temporal dependencies is critical (Bagnall et al., 2018). Many applications, including solar flare prediction (Bloomfield et al., 2012; Bobra & Couvidat, 2015) and seizure detection, require identifying rare but critical events. These tasks suffer from extreme class imbalance: minority events are scarce yet essential, leading standard classifiers to favor the majority class and miss rare events (Buda et al., 2018).

Time series classification (TSC) has advanced from DTW-based methods (Berndt & Clifford, 1994) to CNNs and ResNets (Wang et al., 2017; Ismail Fawaz et al., 2019), and more recently to transformers with strong performance on multivariate data (Zerveas et al., 2021; Ismail Fawaz et al., 2020). Yet most approaches assume balanced data; the imbalanced setting remains underexplored. Classical remedies include SMOTE (Chawla et al., 2002), cost-sensitive training (Zhou & Liu, 2006), or focal loss (Lin et al., 2017), but naive oversampling often yields overfitting or unrealistic samples (Forestier et al., 2017).

Generative Adversarial Networks (GANs) (Goodfellow et al., 2020) provide a way to synthesize data. In time series, models like TimeGAN (Yoon et al., 2019) capture dynamics, while conditional GANs (Mirza & Osindero, 2014) enable class-conditioned generation. However, existing GANs usually treat generation and classification separately, lacking explicit mechanisms to enforce discriminative features.

We propose *UniTSGAN*, a unified transformer-based GAN for imbalanced TSC. The generator, optionally pretrained via a self-supervised masking task, learns temporal representations, while the discriminator uses a dual-head design: one head for adversarial discrimination (real vs. fake) and another for class prediction. This joint training encourages the generator to produce minority-class samples that are both realistic and class-consistent, yielding an effective classifier alongside high-quality augmentation.

To assess synthetic utility, we introduce a downstream evaluation protocol: synthetic minority samples are injected into a balanced training set, and improvements in an LSTM classifier's skill (TSS, HSS2, and their combined DtP score (Bloomfield et al., 2012; Bobra & Couvidat, 2015)) indicate generative quality.

Experiments on seven datasets (six from UCR/UEA (Bagnall et al., 2018) and the SWAN-SF solar flare dataset (Angryk et al., 2020)) show that UniTSGAN consistently outperforms cost-sensitive, oversampling, and deep baseline methods, achieving the best normalized DtP across tasks. It also surpasses recent time series generators under our classification-boost evaluation.

Our contributions are: (1) A unified transformer-based GAN framework for joint time series generation and classification under extreme imbalance; (2) A dual-head discriminator that enforces both authenticity and class fidelity; (3) An evaluation protocol for synthetic data based on downstream classification gains; (4) Extensive experiments and ablations demonstrating state-of-the-art performance.

## 2 RELATED WORK

### 2.1 TIME SERIES CLASSIFICATION

Time series classification (TSC) has a rich literature. Early methods often used hand-crafted features with traditional classifiers, e.g., k-nearest neighbors with Dynamic Time Warping (DTW) (Berndt & Clifford, 1994). Modern approaches leverage deep learning: Convolutional Neural Networks (CNNs) and Residual Networks (ResNets) adapted for temporal data have achieved strong results (Wang et al., 2017; Ismail Fawaz et al., 2019). Specialized architectures like InceptionTime also push the state of the art (Ismail Fawaz et al., 2020). More recently, transformer-based models have been applied to TSC, using self-attention to capture long-range dependencies (Zerveas et al., 2021)(Ismail Fawaz et al., 2020). Such models can learn powerful representations, particularly when pretrained on large time series corpora (Zerveas et al., 2021). However, most existing TSC work assumes balanced or mildly imbalanced classes. The problem of classification under extreme imbalance (e.g., 1:10 minority-to-majority) has seen little attention, motivating specialized methods.

### 2.2 TIME SERIES GENERATION

Generating realistic synthetic time series is useful for simulation and augmentation. GANs have been widely used: for example, TimeGAN (Yoon et al., 2019) integrates an autoencoder loss to capture temporal dynamics and latent consistency. Conditional GANs (CGANs) (Mirza & Osindero, 2014) allow generation of sequences conditioned on labels or contexts. Alternative generative models include variational autoencoders, such as TimeVAE (Desai et al.), uses a VAE architecture with interpretable components to synthesize multivariate time series. Recently, diffusion-based models have been applied to time series forecasting and generation (Rasul et al., 2021). Additionally, latent state-space approaches (e.g., LS4 (Zhou et al., 2023)) use continuous-time latent ODEs to model underlying dynamics. These methods can capture uncertainty and non-stationarity, but they are typically evaluated on similarity metrics or forecasting error, not explicitly on classification utility.

### 2.3 IMBALANCED TIME SERIES CLASSIFICATION

Class imbalance is well-studied for static data, with methods like SMOTE (Chawla et al., 2002), cost-sensitive learning (Zhou & Liu, 2006), and specialized loss functions (Lin et al., 2017). In time series, direct adaptations of these techniques exist (e.g., SMOTE for sequential data (Forestier et al., 2017)), but they may not address temporal coherence. Some works focus on data augmentation for imbalanced time series, synthesizing new minority examples (Forestier et al., 2017). Others pro-

pose imbalance-aware architectures or losses for sequence models (Buda et al., 2018). Despite these efforts, severe imbalance in multivariate TSC (especially with limited data) is still largely underexplored. Our work fills this gap by jointly learning to generate minority sequences and classify under imbalance.

# 3 METHODOLOGY

## 3.1 OVERVIEW

UniTSGAN is an adversarial framework with a transformer encoder as the backbone for both the generator $G$ and the discriminator $D$. The key idea is to incorporate class information directly into the adversarial training via a dual-headed discriminator. Figure 1 illustrates the overall design. The generator learns to produce synthetic time series of the minority class, while the discriminator learns both to distinguish real vs. generated samples and to predict class labels.

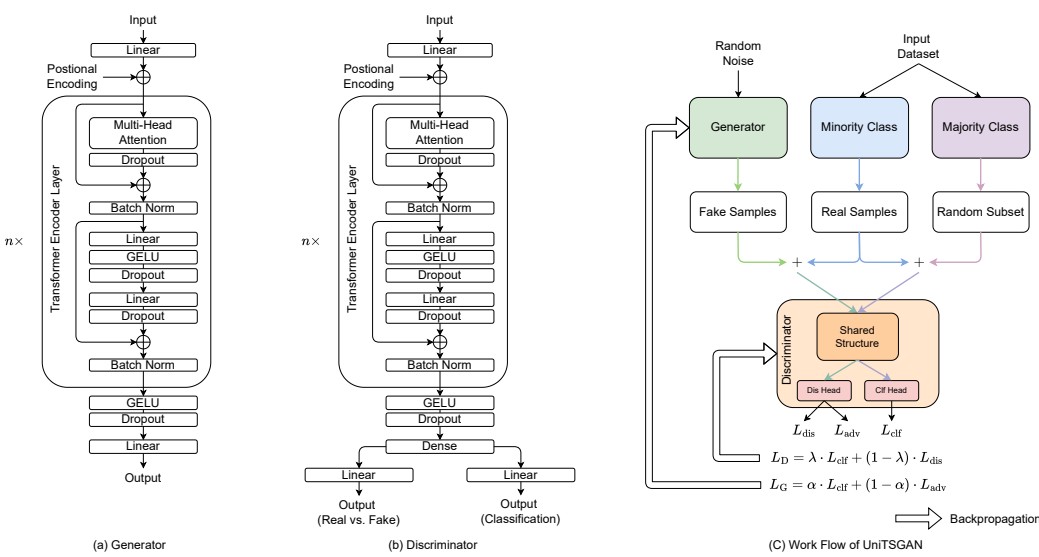

Figure 1: (a) Generator architecture, (b) Discriminator architecture, (c) Work flow of UniTSGAN.

## 3.2 GENERATOR ARCHITECTURE

The generator $G$ is a transformer-based autoencoder. Given a multivariate time series sample $\mathbf{X} \in \mathbb{R}^{l \times m}$ (length $l$, $m$ variables), we first linearly project it to a latent space of dimension $d$:

$$\mathbf{Z} = \mathbf{X}\mathbf{W}_p + \mathbf{b}_p, \quad \mathbf{Z} \in \mathbb{R}^{l \times d},$$

where $\mathbf{W}_p$ and $\mathbf{b}_p$ are trainable. We add learnable positional encodings to $\mathbf{Z}$ to preserve temporal order. The embeddings then pass through $n$ stacked transformer encoder layers. Each layer consists of a multi-head self-attention block (with $h$ heads) followed by a feedforward network with two linear layers (hidden size $d_{\text{ff}}$) and nonlinear activation (we use GELU (Hendrycks & Gimpel, 2016)). Residual connections and layer normalization wrap each sub-block, and we apply dropout (rate 0.1) after attention and in the feedforward block. After the final layer, a linear projection maps the output back to $\mathbb{R}^{l \times m}$, yielding the reconstructed series $\mathbf{X}'$. We note that the generator can be pretrained through unsupervised learning (e.g., (Zerveas et al., 2021)) before adversarial training.

## 3.3 DISCRIMINATOR ARCHITECTURE

The discriminator $D$ also uses a transformer encoder (with the same configuration as $G$) to process input series. However, unlike a standard GAN discriminator, $D$ has a *dual-head* output: one head ($D_{\text{dis}}$) performs binary discrimination (real vs. fake), and the other head ($D_{\text{clf}}$) performs classification (majority vs. minority). Concretely, an input $\mathbf{X}$ is passed through the transformer's layers to

produce an encoded representation. This output is then flattened and fed into a shared linear layer that produces a latent vector. This latent vector is fed into two separate linear heads: one sigmoid unit for $D_{\text{dis}}(\mathbf{X})$, and one sigmoid unit for $D_{\text{clf}}(\mathbf{X})$.

The dual-head design explicitly separates the adversarial and classification tasks. In a vanilla GAN (Goodfellow et al., 2020), the discriminator is oblivious to class identity, which can lead to mode collapse or synthetic samples that do not respect class boundaries. CGANs (Mirza & Osindero, 2014) introduce class conditioning, but they usually treat it as just another input and do not enforce a dedicated classification loss. In contrast, our $D$ learns both to detect fake signals and to extract class-specific features. Thus during training, the classification head encourages $D$ to pay attention to the subtle patterns that distinguish minority from majority sequences, while the discrimination head enforces realism. This multi-task setup improves generalization: the model sees real minority, real majority, and generated minority samples in each batch, so it learns a richer feature space.

### 3.4 Training Objectives

We train $D$ and $G$ in an alternating fashion. Let $x_{\text{min}}$ denote a batch of real minority-class samples, and $x_{\text{maj}}$ an equal-sized batch of real majority-class samples. Let $z \sim \mathcal{N}(0, I)$ be Gaussian noise vectors for the generator. The discriminator loss $L_D$ combines two parts:

$$L_{\text{dis}} = \text{BCE}(D_{\text{dis}}(x_{\text{min}}), 1) + \text{BCE}(D_{\text{dis}}(G(z)), 0), \tag{1}$$
$$L_{\text{clf}} = \text{BCE}(D_{\text{clf}}(x), y), \tag{2}$$

where $\text{BCE}(\cdot, \cdot)$ is the binary cross-entropy loss, $x$ is a balanced batch containing both $x_{\text{min}}$ and $x_{\text{maj}}$, and $y \in \{0,1\}^N$ are the corresponding labels (1 for minority samples, 0 for majority). The total discriminator loss is

$$L_D = (1 - \lambda)L_{\text{dis}} + \lambda L_{\text{clf}}, \tag{3}$$

where $\lambda \in [0, 1]$ balances the adversarial and classification objectives. In practice, we tune $\lambda$ to give appropriate weight to the class loss.

For the generator, we want to fool the discriminator and also ensure the synthetic samples conform to the minority class characteristics. We thus use a combined loss:

$$L_{\text{adv}} = \text{BCE}(D_{\text{dis}}(G(z)), 1), \tag{4}$$
$$L_G = (1 - \alpha) L_{\text{adv}} + \alpha L_{\text{clf}}, \tag{5}$$

where $\alpha \in [0, 1]$ is a hyperparameter. In effect, $G$ is trained to both fool $D_{\text{dis}}$ and produce sequences that $D_{\text{clf}}$ will label as minority. We use separate learning rates and Adam optimizers for $D$ and $G$. This setup gives us flexibility: by adjusting $\alpha$ and $\lambda$, we can prioritize more realistic generation or stronger class discrimination as needed.

## 4 Experiments

### 4.1 Datasets and Preprocessing

We evaluate on seven datasets covering diverse domains: six from the UEA/UCR Time Series Archive (Bagnall et al., 2018) and the SWAN-SF solar flare dataset (Angryk et al., 2020). The UCR/UEA archive provides standard splits and has many labeled time series. To simulate extreme imbalance, for each multi-class dataset we merge its classes into two categories (majority and minority) at a 10:1 ratio. We ensure exactly 10% of samples are positive class. The SWAN-SF dataset is a space weather time series with a naturally severe imbalance; we use the partition scheme of Wen & Angryk (2024). Table 1 summarizes the binary setups. Note that the SWAN-SF tests use four different partitions of the data, as in prior work.

Each dataset is preprocessed by z-score normalization per channel. For the UCR datasets, we use the provided train/test splits. For SWAN-SF, we train on partition 1 and test on partitions 2–5, following Wen & Angryk (2024). During training, all models see only the binary labels (minority vs. majority). For our model, we pretrain the generator on only the minority-class training samples using the masking MSE loss, then proceed with the adversarial training on the combined data.

Table 1: Binary Classification and Imbalance Simulations for Different Datasets

| Dataset | # Classes | # Dimensions | Length | Binary Classification | | Data Size (Simulated) (# 0 + # 1) | |
|---|---|---|---|---|---|---|---|
| | | | | Positive Class (1) | Negative Class (0) | Train | Test |
| Earthquakes | 2 | 1 | 512 | 1 | 0 | 264 + 26 | 104 + 10 |
| EthanolConcentration | 4 | 3 | 1751 | E40, E45 | E35, E38 | 130 + 13 | 132 + 13 |
| FaceDetection | 2 | 144 | 62 | 1 | 0 | 2945 + 294 | 1762 + 176 |
| PEMS-SF | 7 | 963 | 144 | 1, 2, 3, 4 | 5, 6, 7 | 123 + 12 | 69 + 6 |
| SelfRegulationSCP2 | 2 | 7 | 1152 | Positivity | Negativity | 100 + 10 | 90 + 9 |
| SWAN-SF | 5 | 4 | 60 | M, X | FQ, B, C | 87156 + 1464 (P1) | 72238 + 1254 (P2) 41086 + 1424 (P3) 50906 + 1165 (P4) 74365 + 990 (P5) |
| Tiselac | 9 | 10 | 23 | 8 | 1, 2, 3, 4, 5, 6, 7, 9 | 16000 + 1600 | 1540 + 154 |

## 4.2 EVALUATION METRICS

To evaluate classification performance on rare events, we use the True Skill Statistic (TSS) (Bloom-field et al., 2012) and the Heidke Skill Score (HSS2) (Bobra & Couvidat, 2015). Let TP, TN, FP, FN denote the confusion matrix. Then

$$\text{TSS} = \frac{\text{TP}}{\text{TP} + \text{FN}} - \frac{\text{FP}}{\text{FP} + \text{TN}}, \tag{6}$$

$$\text{HSS2} = \frac{2(\text{TP\,TN} - \text{FP\,FN})}{(\text{TP} + \text{FN})(\text{FN} + \text{TN}) + (\text{TP} + \text{FP})(\text{FP} + \text{TN})}. \tag{7}$$

Both range from $-1$ (very bad) to $+1$ (perfect), with 0 indicating no skill. Since each emphasizes different aspects and it is sometimes difficult to determine the better one with two metrics, we adapt DtP (Wen & Angryk, 2024) that combines them into a single scalar by treating $(\text{TSS}, \text{HSS2})$ as a point and measuring its Euclidean distance to the perfect point $(1, 1)$. Specifically,

$$\text{DtP} = \sqrt{(1 - \text{TSS})^2 + (1 - \text{HSS2})^2},$$

and we normalize this to $[0, 1]$ by

$$\text{DtP}_n = 1 - \frac{\text{DtP}}{\sqrt{2}},$$

so that higher $\text{DtP}_n$ means better performance (1 is perfect).

For generative evaluation, we use a post-hoc classification test. As shown in Figure 2, we train a simple 2-layer LSTM on each dataset in two ways: (a) a baseline where we balance classes by dupli-cating minority samples and downsampling the majority; (b) a test condition where we replace the duplicated minority samples with the same number of synthetic samples from a generation model. All other training settings, including architecture, epochs, hyperparameters, and weight initializa-tions, are identical. We then compare the best test $\text{DtP}_n$ achieved by the LSTM in each case. A larger improvement when using synthetic data indicates better quality of the generated minority samples.

## 4.3 BASELINE METHODS

For classification, we compare UniTSGAN's discriminator (applied to test data) against several strong baselines: transformer-based TST (Zerveas et al., 2021), InceptionTime (Ismail Fawaz et al., 2020), Omni-scale CNN (OS-CNN) (Tang et al., 2021), ResNet (Wang et al., 2017), and MLSTM-FCN (Karim et al., 2019). These baselines are trained on the same imbalanced training data (no oversampling) and their best test performance is reported. For data augmentation, we compare our generator against RNN (Elman, 1990), VAE (Kingma & Welling, 2014), LSTM (Hochreiter & Schmidhuber, 1997), and CGAN (Mirza & Osindero, 2014), using the same classification evalua-tion. All methods are tuned to their recommended settings.

## 4.4 RESULTS: CLASSIFICATION

Table 2 reports the mean $\text{DtP}_n$ score of each model on each dataset. Our UniTSGAN (using the discriminator's classifier on test data) achieves the highest average $\text{DtP}_n$ (0.748) and ranks first on

Table 2: Time Series Classification Performance of Different Models on Different Datasets

| Dataset | PreTSGAN | TST (pretrained) | TST (sup. only) | Inception Time | OSCNN | ResNet | MLSTM-FCN |
|---|---|---|---|---|---|---|---|
| Earthquakes | **0.75** | 0.627 | 0.627 | 0.598 | 0.66 | 0.616 | 0.68 |
| EthanolConcentration | **0.698** | 0.573 | 0.561 | 0.636 | 0.589 | 0.636 | 0.581 |
| FaceDetection | **0.6** | 0.581 | 0.58 | 0.572 | 0.541 | 0.54 | 0.538 |
| PEMS-SF | **0.887** | 0.764 | 0.764 | 0.755 | 0.733 | 0.775 | 0.811 |
| SelfRegulationSCP2 | 0.616 | 0.525 | 0.527 | 0.629 | 0.625 | **0.647** | 0.603 |
| SWAN-SF (Avg. over 4 testing partitions) | **0.754** | 0.732 | 0.737 | 0.751 | 0.73 | 0.726 | 0.738 |
| Tiselac | 0.933 | 0.915 | **0.938** | 0.872 | 0.873 | 0.848 | 0.842 |
| **Average DtP$_n$** | **0.748** | 0.674 | 0.676 | 0.688 | 0.679 | 0.684 | 0.685 |
| **Average Rank** | **1.57** | 4.43 | 4.14 | 4 | 4.57 | 4.43 | 4.43 |

**Note**: Bold indicates the best performance and underlining indicates the second best performance.

5 out of 7 datasets. It notably outperforms the supervised-only transformer (TST) on most datasets, demonstrating the benefit of adversarial pretraining. On low-data cases (e.g., EthanolConcentration with only 13 training samples per class), UniTSGAN achieves a $\text{DtP}_n$ of 0.698, which far exceeds the other methods, indicating that synthetic augmentation regularizes and enriches the classifier. Even when not the top performer, UniTSGAN still ranks second on one of them; the only exception is one dataset where transformer-based methods in general perform poorly, likely due to their weaker inductive bias for local patterns. In summary, UniTSGAN yields substantial gains and more robustness in rare-class detection across diverse scenarios.

### 4.5 Results: Generation

Table 3 reports the performance of different generative models under our synthetic-data classification protocol. UniTSGAN achieves the best average downstream performance, with an average normalized distance-to-perfect score (Avg. DtP$_n$) of 0.748. It ranks first on 5 out of 7 datasets and second on the remaining two.

Although certain models (e.g., VAE) obtain comparable average DtP$_n$ values, UniTSGAN attains a substantially lower average rank (1.29), outperforming all other methods. This highlights not only its strong mean performance but also its robustness across diverse datasets. In other words, UniTSGAN consistently generates minority-class samples that are more useful for downstream classification than those produced by alternative generative approaches.

## 5 Conclusion

We have presented UniTSGAN, a unified transformer-based GAN framework for imbalanced time series classification and generation. Our key innovation is a pretrained generator paired with a discriminator that has a separate class-prediction head. This allows the model to produce realistic, minority-class sequences while also learning to classify under imbalance. We introduced a practical downstream-evaluation metric (normalized distance-to-perfect) to jointly assess classification skill and synthetic data utility. Extensive experiments on seven benchmark datasets show that UniTSGAN consistently outperforms or matches state-of-the-art methods in challenging low-data, imbalanced scenarios.

Despite these successes, UniTSGAN still has room for improvement. Currently, the two heads of the discriminator are implemented as simple linear layers; exploring more expressive architectures could enhance both adversarial discrimination and class prediction. Future work could also investigate conditional transformers for the generator, where labels are explicitly incorporated to further regularize synthetic data generation. Finally, going beyond classification, studying the interpretabil-

Table 3: Time Series Generation Evaluation Results

| Dataset | Baseline | PreTSGAN | RNN | VAE | LSTM | CGAN |
|---|---|---|---|---|---|---|
| Earthquakes | 0.705 | **0.764** | 0.752 | 0.759 | 0.752 | 0.752 |
| EthanolConcentration | 0.658 | **0.692** | 0.675 | 0.68 | 0.675 | 0.675 |
| FaceDetection | 0.569 | 0.583 | 0.578 | 0.574 | 0.582 | **0.589** |
| PEMS-SF | 0.92 | **1** | 0.97 | **1** | **1** | **1** |
| SelfRegulationSCP2 | 0.561 | 0.658 | 0.63 | **0.687** | 0.62 | 0.647 |
| SWAN-SF (Avg. over 4 testing partitions) | 0.713 | **0.739** | 0.732 | 0.734 | 0.732 | 0.731 |
| Tiselac | 0.78 | **0.8** | 0.791 | 0.795 | 0.787 | 0.797 |
| **Average DtP$_n$** | 0.701 | **0.748** | 0.733 | 0.747 | 0.735 | 0.742 |
| **Average Rank** | 6 | **1.29** | 3.71 | 2.29 | 3.29 | 2.57 |

**Note**: Bold indicates the best performance and underlining indicates the second best performance.

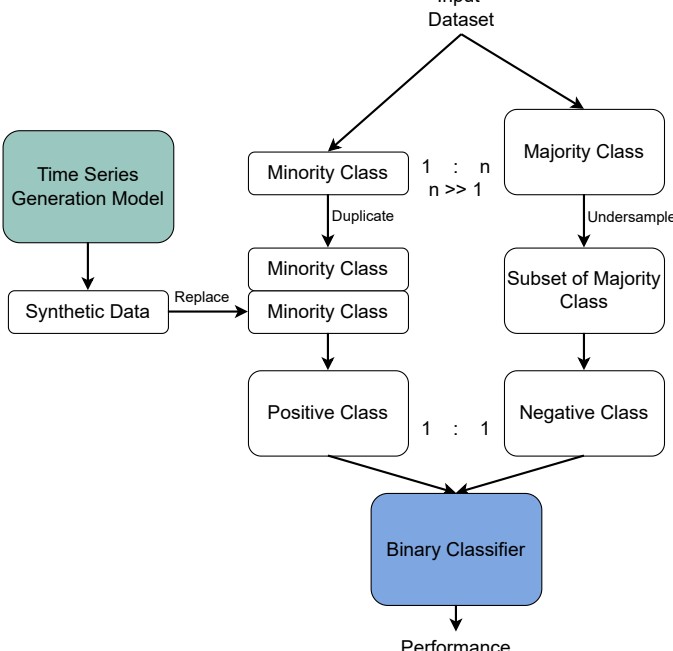

Figure 2: Schematic of the time series generation evaluation protocol. A classifier is first trained on a balanced dataset created by duplicating minority samples and downsampling majority. Then the duplicated minority samples are replaced by synthetic data from a generative model, and the classifier is retrained. Performance gain measures the utility of the synthetic data.

ity of the learned representations and generated sequences may provide valuable insights into the temporal dynamics of rare events.

ACKNOWLEDGMENTS

The authors would like to acknowledge the use of OpenAI's GPT-5 language model for assistance with grammar and spelling checks during the preparation of this manuscript. All ideas, technical contributions, and writing decisions are solely those of the authors.

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
