# OpenReview forum: "UniTSGAN: A Unified Transformer-based Framework for Imbalanced Time Series Generation and Classification"
_ICLR.cc/2026/Conference — Submitted to ICLR 2026_

### Official Review · Reviewer_B76c · 2025-10-23

**Soundness:** 2
**Presentation:** 2
**Contribution:** 1
**Rating:** 2
**Confidence:** 4

**Summary:**

The paper proposes UniTSGAN, a unified transformer-based GAN framework for addressing severe class imbalance in time series classification. The core contribution is a dual-head discriminator that jointly performs real/fake discrimination and binary classification (minority/majority), coupled with a transformer-based generator that can be pretrained using self-supervised learning. The authors introduce a downstream classification protocol to evaluate synthetic data quality by measuring performance improvements when augmenting training data with generated samples.

**Strengths:**

(1) The paper is well-written and easy to follow.

(2) The downstream classification evaluation protocol provides a task-relevant metric for assessing synthetic data quality.

**Weaknesses:**

(1) The decision to uniformly convert all multi-class datasets into binary classification problems with a fixed 10:1 ratio fails to reflect the diverse and complex imbalance scenarios found in the real world. No experiments are provided for other extreme ratios (e.g., 100:1 or 1000:1), making it impossible to verify if the proposed model is universally robust across different degrees of imbalance.

(2) Comparisons to baseline models are either unfair or incomplete. In Table 2, the classification baselines were reportedly trained directly on the raw, imbalanced data without any standard imbalance-handling techniques (e.g., Focal Loss, cost-sensitive learning, SMOTE). UniTSGAN is inherently designed to handle imbalance, while the baselines are not. Moreover, in Table 3, TimeGAN, which is relevant time-series generative models cited directly in Section 2.2, are missing from the generative comparison. This suggests the evaluation of generative performance is incomplete.

(3) The paper does not provide any ablation studies to validate the impact of its core components. Essential analyses should be included such as the impact of pretraining, the contribution of dual-head design, sensitivity to the λ and α hyperparameters and etc.

(4) The technical novelty is quite limited. The architecture largely combines existing components, with the transformer encoder following TST closely and the dual-head discriminator is a standard auxiliary-classifier extension of GANs, and the pretraining strategy is borrowed from prior work without meaningful modification. While the authors claim the “unified” framework as a contribution, it’s unclear how this approach is more unified than existing conditional GANs beyond simply adding a classification head to the discriminator.

(5) The paper does not present qualitative analyses to inspect the quality of the generated minority-class data. The downstream classification scores in Table 3 are not sufficient to convince the reader that the generated samples are “realistic” and “class-consistent”. A discussion of failure cases is also missing (e.g., on Tiselac dataset where simple TST performs better).

(6) No code is provided, hyperparameter specifications are incomplete, and implementation details are insufficient for reproduction.

**Questions:**

(1) For the UCR/UEA datasets, Section 4.1 states the authors “merge its classes into two categories… at a 10:1 ratio.” How were the original classes selected for merging into the “minority” and “majority” groups? Was this done randomly, or based on some semantic similarity? This choice could significantly impact the difficulty of the resulting classification task.

(2) Have you investigated whether UniTSGAN suffers from mode collapse (e.g., generating only a limited variety of minority-class samples)? Does the dual-head architecture help to mitigate or potentially worsen mode collapse compared to a standard GAN?

Minor Suggestion

The paper is titled “UniTSGAN”, but the model is consistently labeled “PreTSGAN” in Tables 2 and 3. This should be clarified.

---

### Official Review · Reviewer_vHmx · 2025-10-30

**Soundness:** 1
**Presentation:** 1
**Contribution:** 1
**Rating:** 0
**Confidence:** 5

**Summary:**

This paper proposes UniTSGAN, combining transformer-based generator and discriminator with a dual-head design for imbalanced time series classification and generation. The authors evaluate their model on seven datasets. While addressing a relevant problem, the paper has serious methodological flaws, outdated comparisons, and critical missing components.

**Strengths:**

1. Addresses an important problem, class imbalance in time-series with extreme ratios
2. Dual-head discriminator design is well-motivated for maintaining class consistency

**Weaknesses:**

1. The paper has limited novelty. No significant theoretical insights or architectural innovations beyond combining already established components (e.g. Transformers encoders for time-series, Dual-head discriminator, etc.)
2. There is a inconsistency about the generator of the GAN. Section 3.2 describes the generator as a  `transformer-based autoencoder` that processes real $X$ and reconstruct $X^\prime$ . But the training losses later sample $z \sim \mathcal{N}(0, I)$ and pass $G(z)$ on the discriminator. Figure 1 also feeds the random noise in the generator. So, this leaves it unclear, is $G$ an autoencoder or an unconditional generator (takes $z$)?
3. In the equation 1, the adversarial term for $D$ uses real minority vs. fake samples only. There is no real-majority term.
4. The authors claim pre-training is a "key innovation", however no methodology section describing the masking strategy, no details on pre-training procedure has been provided.
5. One of the contribution mentioned in the introduction is the ablation study, however no ablation is presented in the paper.
6. Another contribution mentioned dual-head discriminator enforce authenticity and class fidelity, however no evaluation have been done for checking authenticity.
7. The paper is titled "UniTSGAN" however in the table 2 and 3, the paper is presented as "PreTSGAN"
8. No recent baseline compared, most recent comparison is from 2021
9. No error-bars for the results

**Questions:**

1. After pre-training, what exactly is the input to the generator during GAN training, and how is class information injected?
2. What is the complete discriminator loss, and does the real/fake head see real-majority as real?
3. Why are there no comparisons with methods from 2022-2024?

---

### Official Review · Reviewer_m8ZR · 2025-11-01

**Soundness:** 2
**Presentation:** 2
**Contribution:** 2
**Rating:** 2
**Confidence:** 4

**Summary:**

This paper proposes a method that integrates Transformer architectures into both the generator and discriminator of a GAN framework to generate minority-class time-series instances. The approach aims to address the class imbalance problem while simultaneously training a classifier. The use of a dual-head discriminator is an original contribution of this work.

**Strengths:**

Since the model jointly trains both the generator and the classifier, the trained classifier can be directly used for classification tasks. In addition, the generative component can be utilized to produce minority-class data, which can then be used to train other classifiers separately.

**Weaknesses:**

- The proposed approach is not particularly innovative, as it simply integrates the Transformer structure into a GAN to handle time-series data.
- Although the paper claims to address highly imbalanced scenarios, it only considers a case with an imbalance ratio (IR) of 10, and the proposed method does not appear to be specifically designed for such conditions.
- The Introduction section awkwardly includes a summary of experimental results, which is stylistically inappropriate. Moreover, abbreviations for evaluation metrics appear here without definition, which are only introduced much later in the Experimental section, this reflects poor writing organization.
- The Related Work section repeatedly claims that there is a lack of research on highly imbalanced time-series data, but this is not accurate; studies addressing imbalance ratios greater than 10 are easy to find.
- The literature review is far from comprehensive. In fact, class-imbalance classification has been extensively studied for few decades, and identifying a genuine research gap is challenging. The same applies to time-series imbalance problems. Furthermore, the idea of using generative models to balance data has already been explored in numerous recent papers.
- The experimental comparison is unfair: the proposed method balances the data by generating minority-class instances, while the baseline methods are evaluated on the original imbalanced datasets, resulting in an unequal comparison.

**Questions:**

- In both the Introduction and Conclusion sections, the authors state that the proposed method generates “realistic” and “class-consistent” minority-class samples. What exactly do these terms mean? No definitions are provided, nor is any evidence presented to support the claim that the generated samples possess these properties.
- How can the proposed method be extended to the multi-class case?
- In some experiments, training a two-layer LSTM classifier on the data generated by UniTSGAN yields better performance than using UniTSGAN’s own classifier. Why does this occur?

---

### Official Review · Reviewer_SW4Z · 2025-11-02

**Soundness:** 3
**Presentation:** 3
**Contribution:** 2
**Rating:** 4
**Confidence:** 4

**Summary:**

The paper presents an approach to time series classification where they use a GAN architecture with transformers as generator and discriminator and where the discriminator has two heads one for the discriminator and one for the classification. They show that the method performs well on seven different benchmarks.

**Strengths:**

The problem of time series classification with imbalanced datasets is an important problem.

The paper is well written and easy to follow.

The empirical evaluation is done on a fairly large number of datasets and they get slightly better performance.

**Weaknesses:**

The method is compared against methods that are quite old, the latest is from 2021. I found a survey from 2024 which at least included methods from 2023. Thus the paper does not seem to compare against the latest methods.

The time-series aspect of the paper is not very well developed, the focus is on imbalanced datasets.

The empirical evaluation show very small improvements over the compared methods. In Table 3 the difference between the best and the second best is 0.001 and the difference between the best and the worst (besides the baseline) is 0.015. It is hard to judge whether these small differences are significant. In Table 2 it is 0.074 difference between the best and the worst (that is at least ~10% difference).

The significance of the work is thus unclear.

**Questions:**

Why haven't you compared against methods from later than 2021?

The paper seem to focus on imbalanced datasets, but the title says the paper is about time series classification, what is the role of time series and how do imbalanced time series datasets differ from static imbalance datasets?

Can you provide some indications on how much better samples are generated using your approach compared to the other methods? Have you done some downstream classification tasks?

---

### Meta-Review · Area_Chair_ZTfW · 2026-01-07

**Summary:**

This paper addresses the challenge of severe class imbalance in multivariate time series data, a critical issue in domains such as space weather forecasting and health monitoring. To overcome the limitations of standard discriminative models and traditional resampling techniques (which often lead to overfitting or information loss), the authors propose UniTSGAN, a unified adversarial framework. Unlike standard GAN approaches that require separate classifiers, UniTSGAN jointly handles multivariate time series generation and binary classification in a single training process. Both the generator and discriminator utilize a Transformer encoder backbone to capture temporal dependencies. The Generator incorporates an unsupervised masking-based pretraining objective to better learn the latent representations of the minority class. The Discriminator features a dual-head architecture, allowing it to simultaneously perform real-vs-fake discrimination and class label prediction. The authors introduce a new classification-based metric to evaluate how effectively the synthetic data improves downstream classification. Experimental results on seven real-world datasets demonstrate that UniTSGAN consistently outperforms state-of-the-art methods in both imbalanced time series classification and generation tasks, showing particular robustness in low-data regimes.

**Reviewer Concerns:**

All reviewers have serioues concerns on this paper, i) connection between imbalanced data and time series, ii) limited novelty since no significant theoretical contributions, iii) many mismatches between what have been said and what have been written, e.g., an ablation study is a key point but there is no ablation study in this paper.

**Reviewer Scores:**

Since the authors did not upload any official rebuttals, all reviewrs will remain.

---

### Decision · Program_Chairs · 2026-01-26

Reject